# A Generic Sequential Stimulation Adapter for Reducing Muscle Fatigue during Functional Electrical Stimulation

**DOI:** 10.3390/s21217248

**Published:** 2021-10-30

**Authors:** Gongkai Ye, Saima S. Ali, Austin J. Bergquist, Milos R. Popovic, Kei Masani

**Affiliations:** 1KITE-Toronto Rehabilitation Institute, University Health Network, Toronto, ON M4G 3V9, Canada; jerry.ye@mail.utoronto.ca (G.Y.); austinb@ualberta.ca (A.J.B.); milos.popovic@uhn.ca (M.R.P.); 2Department of Electrical and Computer Engineering, University of Toronto, Toronto, ON M5S 3G4, Canada; saiima.ali@mail.utoronto.ca; 3Institute of Biomedical Engineering, University of Toronto, Toronto, ON M5S 3G9, Canada

**Keywords:** sequential stimulation, functional electrical stimulation, rehabilitation, fatigue reduction

## Abstract

Background: Clinical applications of conventional functional electrical stimulation (FES) administered via a single electrode are limited by rapid onset neuromuscular fatigue. “Sequential” (SEQ) stimulation, involving the rotation of pulses between multiple active electrodes, has been shown to reduce fatigue compared to conventional FES. However, there has been limited adoption of SEQ in research and clinical settings. Methods: The SEQ adapter is a small, battery-powered device that transforms the output of any commercially available electrical stimulator into SEQ stimulation. We examined the output of the adaptor across a range of clinically relevant stimulation pulse parameters to verify the signal integrity preservation ability of the SEQ adapter. Pulse frequency, amplitude, and duration were varied across discrete states between 4 and 200 Hz, 10 and100 mA, and 50 and 2000 μs, respectively. Results: A total of 420 trials were conducted, with 80 stimulation pulses per trial. The SEQ adapter demonstrated excellent preservation of signal integrity, matching the pulse characteristics of the originating stimulator within 1% error. The SEQ adapter operates as expected at pulse frequencies up to 160 Hz, failing at a frequency of 200 Hz. Conclusion: The SEQ adapter represents an effective and low-cost solution to increase the utilization of SEQ in existing rehabilitation paradigms.

## 1. Introduction

During voluntary muscle contractions, motor units are recruited asynchronously with respect to each other. This asynchronous motor unit recruitment allows for fused contractions of relatively high strength to be achieved at relatively low individual motor unit discharge rates (e.g., <20 Hz) due to the temporal summation of individual twitch responses [1]. In contrast, conventional FES generates muscle contractions predominantly by the synchronous depolarization of motor axons beneath the stimulating electrodes [2]. Such synchronous motor unit recruitment means that stimulation frequencies in the range of 35–60 Hz [3,4] are required to generate contractions that are fused and that are of functionally relevant strengths. The relatively high motor unit discharge rates associated with conventional FES increases the metabolic demand placed on recruited motor units, contributing to greater levels of neuromuscular fatigue compared to voluntary contractions [5,6,7,8].

To prevent rapid neuromuscular fatigue associated with FES-evoked contractions, researchers have developed a promising FES technique which mimics the asynchronous motor unit recruitment pattern that occurs during voluntary contractions. This technique involves the “sequential” rotation of stimulation pulses between multiple active electrodes positioned over the muscle belly (SEQ). This is driven by the rationale that each SEQ electrode can be activated at a low “average” frequency (e.g., 10 Hz stimulation delivered to and sequentially distributed among four active electrodes), while maintaining a high composite frequency delivered to the muscle, or muscle group, as a whole (e.g., 40 Hz). If each SEQ electrode activates distinct motor unit populations, as has been shown for the plantarflexors [9], then the SEQ technique will reduce motor unit discharge rates and subsequent fatigue compared to conventional FES.

To date, a number of SEQ techniques have been employed and have demonstrated reduced neuromuscular fatigue in human studies [9,10,11,12,13,14,15,16,17,18,19]. In initial SEQ stimulation studies, pulse trains were rotated between multiple active electrodes, described as “sequential” or “cyclical” [18] and “alternating” [19] stimulation. In one of these studies [18], it was found that reducing the time that an electrode was stimulated before switching to the next electrode (i.e., reducing the duty cycle by reducing the number of pulses in a train) resulted in less fatigue. This work motivated the use of a different technique whereby individual pulses, rather than pulse trains, were interleaved between multiple active electrodes, described as “asynchronous” [10,11,12,13,15], “distributed” [15,17], and “spatially distributed sequential” [9,14,16] stimulation. To date, this interleaved SEQ technique has been shown to reduce fatigue of the human finger flexors [15], plantarflexors [9,14,16], dorsiflexors [14], knee flexors [14], and knee extensors [10,11,12,13,14,17].

Despite the consistent demonstration of the effectiveness of SEQ in reducing neuromuscular fatigue compared to conventional FES, there has been limited uptake of the technique by both the research community and clinical practices. Currently, SEQ delivery is typically accomplished by programming four individual stimulator channels at a quarter of the desired composite frequency, 90° phase shifted from each other. However, stimulators may not possess the capability to program SEQ protocols, and stimulators that can deliver SEQ require the user to possess programming proficiency. These are limitations to the adoption of SEQ.

In this work, we propose the use of a small, battery-powered electronic device, called an SEQ adapter, to transform the output of any commercial FES stimulator to SEQ stimulation. The SEQ adapter receives electrical pulses from the output of a single stimulation channel and sequentially rotates the delivery of those pulses, pulse-by-pulse, among four separate outputs instead of programming four separate channels. Using this SEQ adapter, we previously demonstrated that using it to apply SEQ stimulation reduced the neuromuscular fatigue of quadriceps femoris muscle contractions compared to conventional FES, and that each SEQ electrode activated relatively distinct motor unit populations [20]. These results suggest the SEQ adapter can be used in clinical settings to reap the benefits without the overhead of needing to purchase new stimulators or the overhead of programming existing ones, potentially enabling SEQ to become more widely adopted within research communities and clinical practices.

Presently, the device is in a prototyping phase. We present a design of the device, as well as a preliminary assessment of the signal integrity of the SEQ adapter output compared to the stimulator output, across a range of clinically relevant stimulation parameters.

## 2. Materials and Methods

The SEQ adapter (Tecnalia R&I Spain, San Sebastian, Spain) is shown in Figure 1. The adapter receives cathodic input from a single stimulator channel via a standard 2 mm female connector which is routed to the four cathode electrodes. The anode cable is directly attached to the anode electrode. The pulse detection circuit transforms input current pulses into digital logic sent to a reduced instruction set computing microcontroller (ATtiny44A; ATMEL, San Jose, CA, USA). The digital logic controls the multiplexer output to rotate the stimulation pulses across output channels. The microcontroller performs all logical operations including control of the switching circuit and light-emitting diode (LED) indicator. The SEQ adapter measures the current flow at the output and a comparator activates channel switching when this current level surpasses a set minimum. There is always one channel active. There is also a timer that prevents switching within 10ms of the previous channel switch. During the multiplexing of stimulation pulses, the LED flashes with one eighth of the input frequency. Power supply of the adapter is provided by a single rechargeable lithium polymer battery, and it is recharged via a micro-USB port.

The performance of the adapter was tested by varying the following parameters: pulse amplitude, pulse duration, and pulse frequency. The selected values for each parameter were based on ranges of clinically relevant settings and are summarized in Table 1. Every combination of parameter values was tested. With 10 pulse amplitudes, 6 pulse durations, and 7 frequencies, a total of 420 combinations of parameters were tested. Each of these combinations will be called a “trial”, with each trial comprised of a train of 80 pulses. Each trial was conducted twice: once for the SEQ adapter and once for the originating stimulator.

Rectangular monophasic electrical pulses were delivered from a constant-current stimulator (DS7A; Digitimer, Welwyn Garden City, UK) that was controlled by LabChart software (PowerLab 16/35; ADInstruments Pty Ltd., New South Wales, AU, USA). 

For the adapter trials, the adapter was connected in series with the output of the stimulator. Each of the 4 adapter outputs were connected in series with 75 Ω resistors, with an output of 20 pulses per channel, per trial. For the stimulator trials, the single stimulator output was connected in series with a 75 Ω resistor with an output of 80 pulses per trial. The voltage across these constant loads was sampled at 110 kHz using LabVIEW software (PCI-6255; National Instruments, Austin, TX, USA) and stored on a computer for subsequent analysis using custom written code with computational software (MATLAB R14b; The Mathworks, Inc., Natick, MA, USA).

Pulse amplitude was calculated as the zero-to-peak voltage divided by the resistance in series and expressed in mA. Pulse duration was calculated as the length of time (sampling rate divided by the number of samples) between the mid-crossings of the descending and ascending transitions of each negative-polarity pulse (cathodal) and expressed in μs. Pulse frequency was calculated as 1 divided by the length of time between the mid-crossings of the descending transitions of each negative-polarity pulse and the next negative going pulse, expressed in Hz. Lastly, the performance of the adapter with respect to each pulse parameter (pulse amplitude, duration, and frequency) was measured by % Error, calculated as the difference between stimulator output and adapter output (averaged across 4 adapter channels) and expressed as a percentage of stimulator output.

## 3. Results

### 3.1. Qualitative Overview

Figure 2 shows stimulation pulses recorded from stimulator and adapter outputs across a range of pulse amplitudes (Figure 2A,B) and pulse durations (Figure 2C,D). Stimulation pulses from adapter channels 2, 3, and 4 (CH2, CH3, and CH4, respectively) were omitted for illustrative purposes. Qualitatively, CH1 of the adapter output closely replicates the stimulator output across each pulse amplitude (10 to 100 mA in 10 mA steps in Figure 2A vs. Figure 2B) and pulse duration (50, 100, 200, 500, 1000, and 2000 μs in Figure 2C vs. Figure 2D). Figure 2E,F show stimulation pulses recorded from the stimulator and all 4 adapter channels at 160 Hz and 200 Hz input frequencies, respectively. The adapter is effective at sequentially rotating pulses among all 4 adapter channels at frequencies up to 160 Hz (Figure 2E), resulting in a 40 Hz frequency delivered at each output channel. In contrast, Figure 2F shows that the adapter fails to sequentially rotate pulses among all 4 adapter channels at an input frequency of 200 Hz. Rather, all pulses are delivered to CH1 at 200 Hz. The device also failed at frequencies above 200 Hz, although that data are omitted. 

### 3.2. Quantitative Performance

In each of the quantitative data points for the SEQ adapter, the parameter value is calculated as an average across all pulses in the train, i.e., across all channels. Figure 3 shows XY scatter plots depicting pulse amplitude data across a range of pulse durations, separated into panels by pulse frequency. Figure 4 shows XY scatter plots depicting pulse duration data across a range of pulse amplitudes, separated into panels by pulse frequency. Figure 5 shows XY scatter plots depicting pulse amplitude data across a range of pulse frequencies, separated into panels by pulse duration. Data for 200 Hz are omitted from each figure due to failure of the adapter to sequentially rotate pulses (Figure 2F). In each figure, notice that all data lie narrowly along the dashed line of unity in each panel, resulting in % Error values less than 1% (see panel insets).

## 4. Discussion

The performance of the SEQ adapter was excellent across a range of physiologically relevant pulse amplitudes (10–100 mA), pulse durations (50–2000 μs), and pulse frequencies (4–160 Hz), with outputs tracking the output of the stimulator for each pulse parameter within 1% error. Physiologically, a difference of less than 1% between stimulator and adapter outputs is expected to have little to no effect on motor unit recruitment patterns, but this should be further verified.

The SEQ adapter failed to perform at an input frequency of 200 Hz (i.e., 50 Hz pulse frequency at all 4 adapter output channels). However, considering that the rationale for delivering SEQ stimulation is to reduce motor unit discharge rates and subsequent fatigue by delivering stimulation at physiologically low pulse frequencies (e.g., <20 Hz), the failure mode of the SEQ adapter at 200 Hz does not invalidate its use for the suggested clinical application. An output frequency of 50 Hz is consistent with stimulation frequencies utilized during conventional FES [3,4] and as such, even if the SEQ adapter were to perform at an input frequency of 200 Hz, it is likely to be ineffective for reducing neuromuscular fatigue compared to conventional FES. 

Despite these promising results using monophasic pulses from a common electrical stimulator, it will be important to test the performance of the SEQ adapter when delivering biphasic pulses which are also common in many FES applications. As the critical feature of electrical pulses in its function is the negative polarity that was appropriately transferred using the SEQ adapter, we are confident that the SEQ adapter works properly when used with biphasic pulses. However, the positive-polarity anodal deflection, which is a minor feature of electrical pulse, is somewhat truncated at the adapter outputs compared to the stimulator output although not quantified presently (Figure 2). Although this should have little effect on motor unit recruitment patterns since the positive-polarity anodal deflection is responsible only for hyperpolarization, this may lead to premature cathodal corrosion related failures at the adapter outputs [21]. Further testing with biphasic pulses is necessary and analysis of output performance under physiological loads is also necessary and suggested as future work. We measured the stimulator output separately from the adapter output because it was impossible to measure the outputs at the same time without losing the current. Therefore, our results are subject to an assumption that the stimulator output must be the same in the two conditions and this is a possible limitation.

While this SEQ adapter is not yet commercially available, it represents what could be a relatively inexpensive and user-friendly option for researchers and clinicians alike to deliver SEQ stimulation seamlessly with any commercially available stimulator. One disadvantage of the current setup is that many more wires are required, which may be a nuisance to the user. In future iterations of the device, it would be possible to integrate the electrode array with the adapter as a single unit, reducing excess wires. Such an adapter could increase the utilization of this promising FES technique, enhancing the effectiveness of FES therapy in a clinical setting.

## 5. Conclusions

SEQ stimulation is a promising alternative to traditional stimulation protocols. However, most stimulators are limited in their ability to deliver SEQ. Our SEQ adapter prototype was able to effectively transform traditional stimulation into SEQ stimulation with less than 1% error in output stimulation parameters. Further investigation of the SEQ adapter performing under biphasic stimulation conditions and physiological load are required. However, our results present a glimpse of a low cost, backward compatible option that would promote the adoption of SEQ in current FES therapies.

## Figures and Tables

**Figure 1 sensors-21-07248-f001:**
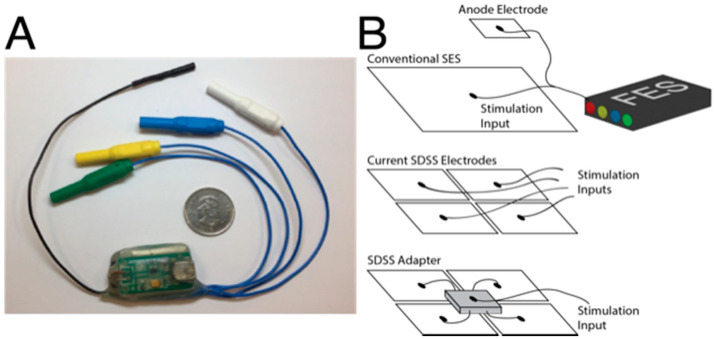
Sequential stimulation adapter prototype (**A**). SDSS setups without an adapter requires four input channels from the stimulator whereas a SDSS adapter requires only one input channel (**B**).

**Figure 2 sensors-21-07248-f002:**
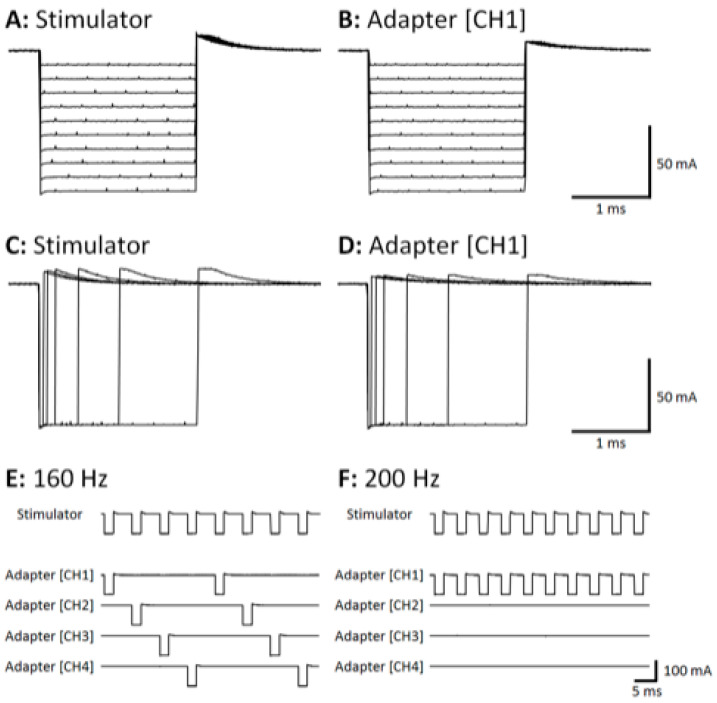
CH1, channel 1. Sample of pulses used to test the sequential stimulation (SEQ) adapter (**A**,**C**). Varying pulse amplitude with constant pulse duration (**A**), and varying pulse duration with constant pulse amplitude (**C**) are shown. Data were collected in separate trials but overlaid for visualization purposes. Sample of output pulse from a single channel of the SEQ adapter (**B**,**D**) corresponding to pulse settings in (**A**,**C**), respectively. Successfully rotated pulses from SEQ (**E**). Demonstrated failure at 200 Hz (**F**).

**Figure 3 sensors-21-07248-f003:**
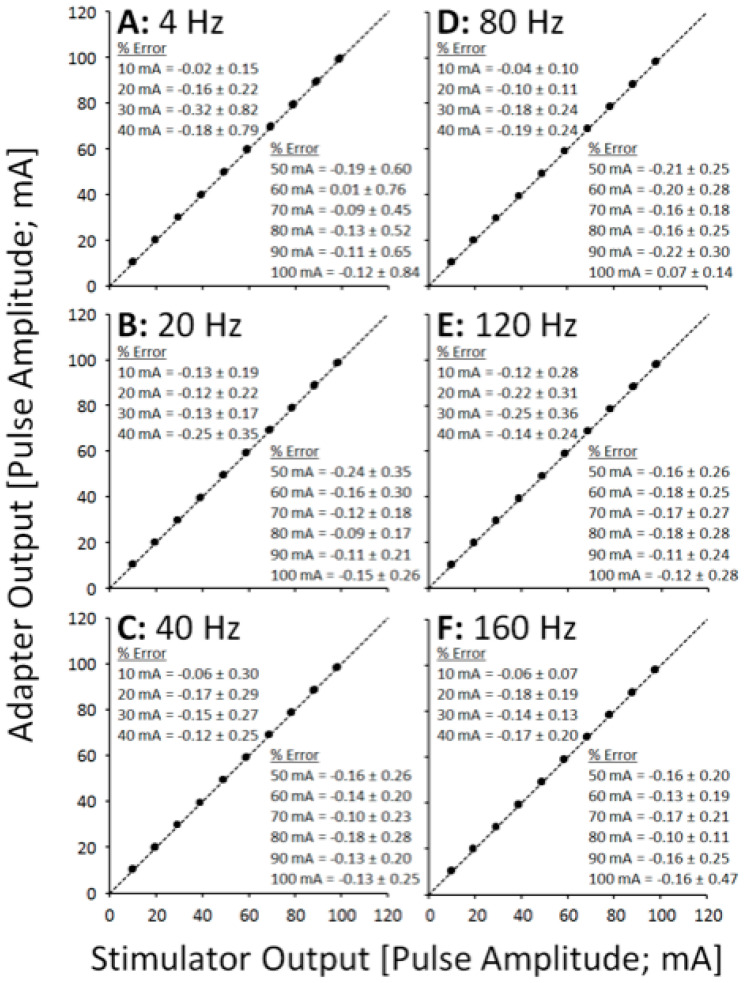
Scatter plots showing adapter pulse amplitude outputs vs. stimulator pulse amplitude outputs across frequencies (**A**–**F**).

**Figure 4 sensors-21-07248-f004:**
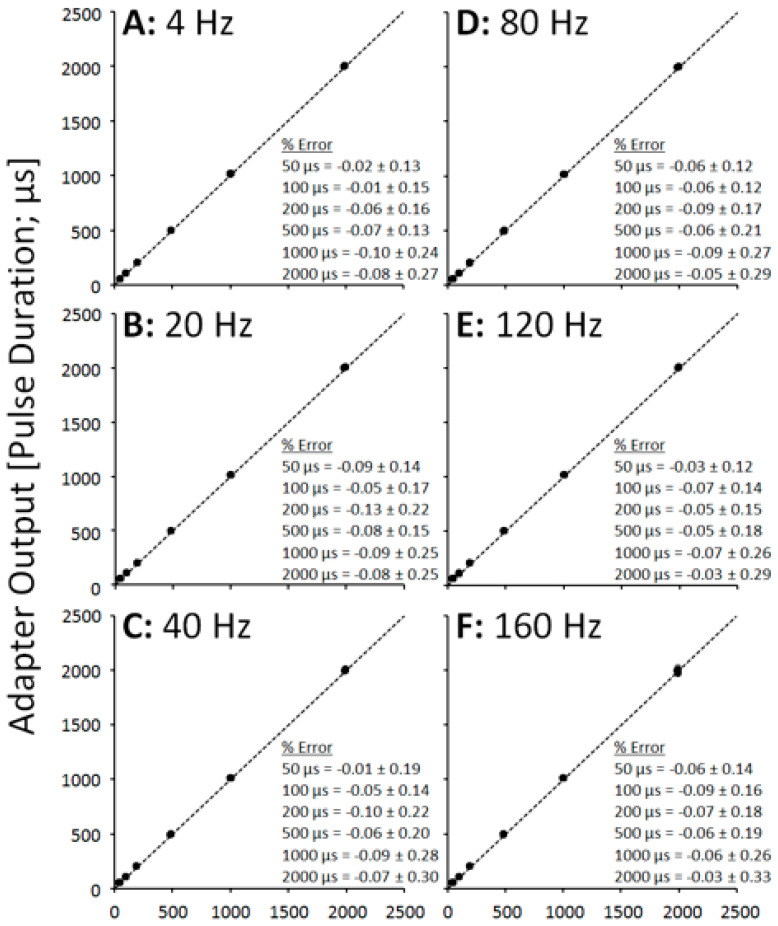
Scatter plots showing adapter pulse duration outputs vs. stimulator pulse duration outputs across frequencies (**A**–**F**).

**Figure 5 sensors-21-07248-f005:**
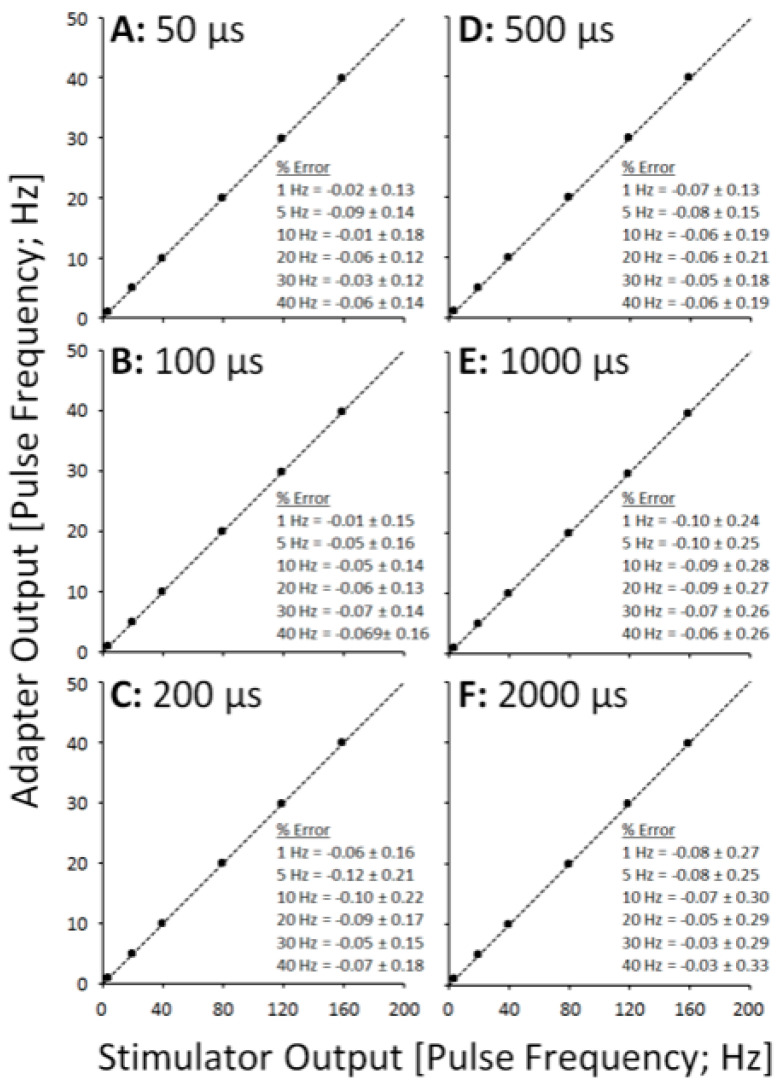
Scatter plots showing adapter pulse frequency outputs vs. stimulator pulse frequency outputs across pulse durations (**A**–**F**).

**Table 1 sensors-21-07248-t001:** Parameters used to test the SEQ adapter. Every combination of parameters was tested in individual trials, for a total of 420 trials.

Parameter	Settings
Pulse Frequency (Hz)	4, 20, 40, 80, 120, 160, 200
Pulse Amplitude (mA)	10, 20, 30, 40, 50, 60, 70, 80, 90, 100
Pulse Duration (μs)	50, 100, 200, 500, 1000, 2000

## Data Availability

Data available on request from the corresponding author.

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
