# Peer review of "A Generic Sequential Stimulation Adapter for Reducing Muscle Fatigue during Functional Electrical Stimulation"

_sensors, 2021, doi:10.3390/s21217248_

Round 1

Reviewer 1 Report

In this paper, the authors present some charactarisation of a useful little device for delivering FES. While the paper is well written and from the data presented the device itself appears to work and will certainly be of use to the FES community, I found the degree of detail and degree of charactaristion to be lacking, and worthy of expanding. 

The authors test a range of FES parameters and show that their multiplexer faithfully outputs monopolar input pulses for input pulse trains up to 200 Hz. What are the other limits to the performance of their device? e.g. what is the smallest / largest amplitude pulses which will be transmitted faithfully? What is the shortest and longest pulse-width which will be transmitted faithfully? Knowing all (not just some) of the limits of performance for a device will be useful for knowing whether that device can be applied in my lab, given the protocols I use. The search for these limits need not be exhaustive in terms of testing every possible parameter combination. 

Can the authors elaborate more as to why the device fails at 200 Hz? From the way the paper is written, it is ambiguous as to whether 200 Hz is the upper frequency limit or whether higher-frequency pulses will be multiplexed correctly (which is implied by the language "a failure mode at 200 Hz"). This should be made clear. 

Similarly, the authors mention that they have not yet tested biphasic or anodic pulses and acknowledge this in the discussion. As biphasic waveforms are recommended for many FES applications (in particular recruitment of small muscle groups ) it would be important to know whether their device functions for anodic pulses. If it is simply less accurate in terms of total current, that is ok if the stimulator can be programmed to generate the desired outcome as long as the stimulus is repeatable. If it does not transmit anodic pulses at all, that is more of a problem. 

Some other comments... 

In figue 1, a block diagram is referenced but not provided. 

Why was the test resistance of 75 ohms chosen? Why no reactive component given capacitance of skin in real FES applications? 

How it it (the device) recharged? From a USB plug or wirelessly? 

Reviewer 2 Report

Overview

The SEQ device is a good idea for reasons explained in the paper. The authors might have added that one of the disadvantages of have many electrodes per muscle is that more inconvenient wires are required which is a nuisance to the user: locally distributing the pulses at the muscle avoids this disadvantage.

Comments

Tests on the device are described but these use monophasic pulses whereas biphasic pulses are preferred because they are safer. The second pulse of a biphasic pair of pulses is sometimes produced actively (often of the same pulse width as the first pulse), but some times it is passive, produced by allowing a series capacitor to discharge. With which of these would the SEQ device work? The paper is unhelpful because there is no specification. Could a specification be added?

The results are presented by comparing the stimulator output to the adaptor output. If the stimulator output is being measured on another channel of the stimulator (not the one with the adaptor), why is this a fair basis for comparison? I can guess the answer but the reason should be stated.

I think the authors should do what they say in the Discussion and test the device with biphasic currents and include those results in this paper.

Round 2

Reviewer 2 Report

I still think that the paper would be improved by a technical specification. Specifically, I would like to understand (i) whether the input cable has one or two wires, and (ii) what causes the internal multiplexor to advance to the next output (Is it the next step-change in current or what?). Will this information be in your follow-on paper? Why would it be better there than here?

Author Response

Response to reviewers

  1. I still think that the paper would be improved by a technical specification. Specifically, I would like to understand (i) whether the input cable has one or two wires, and (ii) what causes the internal multiplexor to advance to the next output (Is it the next step-change in current or what?). Will this information be in your follow-on paper? Why would it be better there than here?

Thank you for the comment. We noticed that the description on the use of adapter is not very clear. One cable from a stimulator channel (cathode cable) goes into the adapter input, while the anode cable directly goes to the anode electrode. The four output cables of the adapter are connected to four cathode electrodes. We added a set of schemes in Figure 1 to illustrate the use of adapter in a better way. We also revised the sentence at L91-92:

“The adapter receives cathodic input from a single stimulator channel via a standard 2 mm female connector which is routed to the four cathode electrodes. The anode cable is directly attached to the anode electrode.”

The adapter we purchased was designed and developed at Tecnalia R&I Spain. They do not intend to disclose the circuit logic inside and the only information that they can share is the description in the manuscript at L92-99. Although we fully understand your question, we are unfortunately not able to clearly answer to your question. We hope that this paper contributes to the field by introducing the idea of SDSS adapter, instead of the technological details.